# De-Labelling Penicillin Allergies in the Paediatric Emergency Department

**DOI:** 10.3390/antibiotics14121222

**Published:** 2025-12-04

**Authors:** Owen Hibberd, Spyridon Karageorgos, Melanie Ranaweera, Patrick Joseph William Mullally, Marius-Valentin Constantin, Aideen Byrne, Michael J. Barrett

**Affiliations:** 1Emergency and Urgent Care Research in Cambridge (EURECA), PACE Section, Department of Medicine, Cambridge University, Cambridge CB2 0QQ, UK; o.hibberd@nhs.net; 2Blizard Institute, Queen Mary University of London, London E1 2AT, UK; s.karageorgos@qmul.ac.uk (S.K.);; 3Pediatric Emergency Department, Aghia Sophia Children’s Hospital, 11527 Athens, Greece; 4Department of Paediatric Emergency Medicine, University Hospital Southampton, Southampton SO16 6YD, UK; 5Pediatric Emergency Department, Children’s Hospital for Wales, Cardiff CF14 4XW, UK; 6Department of Paediatric Allergy, Children’s Health Ireland, D12 N512 Dublin, Ireland; 7Department of Paediatrics, School of Medicine, Trinity College Dublin, D08 W9RT Dublin, Ireland; 8Paediatric Emergency Research and Innovation (PERI), Department of Paediatric Emergency Medicine, Children’s Health Ireland, D12 N512 Dublin, Ireland; 9Women’s and Children’s Health, School of Medicine, University College Dublin, D04 V1W8 Dublin, Ireland

**Keywords:** pediatric emergency, penicillin allergies, de-labeling

## Abstract

While many paediatric patients have a penicillin allergy label, most do not have a true allergy. The penicillin allergy label is associated with a lifetime risk of avoidable use of broad-spectrum antibiotics, higher healthcare costs, and poorer clinical outcomes. In this review, we present different types of penicillin allergies, de-labelling approaches, and significance on paediatric patients. We also discuss parental perspectives regarding penicillin de-labelling in the emergency setting. We highlight that despite the challenges posed by barriers such as overcrowding and the need for quick patient turnover in the PED, the availability of resources and expertise in managing potential allergic reactions makes the PED an ideal environment where PCN de-labelling can be both feasible and effective. We show that further education of both parents and healthcare professionals is essential to overcoming misconceptions, alleviating safety concerns, fostering trust in the de-labelling process, and normalising de-labelling in the PED.

## 1. Introduction

The global prevalence of penicillin allergy (PCN) labels is estimated to be approximately 9% [1]. In total, 75% of these labels are applied in children under three years of age and carried through life [2]. However, oral drug challenges demonstrate that around 95% of children with a reported PCN allergy are in fact penicillin-tolerant [3]. A small proportion of patients with reported allergies have a true allergy [3]. More often than not, ambiguous, suspected, or spurious allergies are commonly reported [4]. Studies have shown that unconfirmed PCN allergy labels are harmful, as penicillin derivatives are often a first-line therapy for common infections [5,6]. The PCN allergy label is linked to increased use of broad-spectrum antibiotics, higher rates of adverse events, greater costs, and worse clinical outcomes [4,7]. This highlights the need for establishing penicillin-de-labelling strategies. The rationale is to

1.Improve the ability to use first-line antibiotics, especially related to patients with a reported beta-lactam allergy.2.Reduce antibiotic-associated adverse events.3.Reduce inpatient length of stay.4.Improve the ability of provider history-taking relevant to antibiotic allergies.5.Save health costs.

In this review, we present types of penicillin allergies, de-labelling approaches, and significance on paediatric patients. We also discuss parental perspectives regarding penicillin de-labelling in the emergency setting.

### 1.1. Types of Penicillin Allergies Reported

Adverse drug reactions (ADRs) can be clinically categorised into type A (on-target) and type B (off-target) reactions [8]. Type A reactions comprise 80% of ADRs, are dose-dependent, and predictable based on the drug’s therapeutic effect [8]. A common reaction in children is antibiotic-associated diarrhoea due to alterations in the bacterial microbiome, whereas type B reactions account for 20% of ADRs and encompass hypersensitivity reactions. Whilst these remain dose-dependent, they are not predictable based on the drug’s therapeutic effect [8]. A significant issue arises, whereby all ADRs are erroneously considered as being allergies (Table 1).

Hypersensitivity reactions (type B reactions) are IgE, IgG, or T lymphocyte immunologically mediated responses to antibiotics in a sensitised individual [9]. The Gel and Coombs system classifies these hypersensitivity ADRs into four types (I, II, III, and IV). Immediate, type I reactions, induced by an IgE-mediated mechanism, occur within the first hour of drug administration and may progress to anaphylaxis [9]. Delayed type IV reactions, which develop more than an hour post drug exposure, are often induced by a T-lymphocyte-dependent reaction and typically manifest as a maculopapular exanthem or more extreme dermatological variations [9].

There are other proposed classification systems based on the patient’s phenotype and endotype, as well as biomarkers, including in vivo, in vitro, and genetic markers [10]. However, these have limited widespread application in clinical practice due to their complexity and time-consuming nature.

Allergy symptoms can be classified as high or low risk (Table 2).

In the vast majority of cases, the label of a PCN allergy is applied to both adult and paediatric patients with low-risk allergy symptoms, with skin rashes being associated with most childhood applied labels [8,11]. Being able to risk-stratify patients using commonly reported symptoms from parents is especially useful within the clinical context. Depending on the associated risk, ongoing recommendations can be provided [12].

### 1.2. Why De-Labelling Penicillin Allergy Is Important

If a child is mis-labelled as allergic to PCN, this label can carry into adulthood, affecting their individual care and having broader implications for antimicrobial stewardship and healthcare costs. Many children are labelled as allergic to PCN due to low-risk symptoms or non-immune-mediated symptoms being reported. This can include instances where parents are unable to recall the allergy [4].

Reassessing the allergy status and removing the label of PCN allergy when it is safe is known as de-labelling. De-labelling can enhance children’s access to first-line antibiotics, as PCN and other beta-lactam antibiotics are the preferred first-line treatments for paediatric infections due to their efficacy, narrow-spectrum coverage, and established safety [13]. This may also help reduce antimicrobial resistance, as second-line broad-spectrum antibiotics, such as macrolides or fluoroquinolones, which are often used in children with a label of PCN allergy, are associated with increased antibiotic resistance. Broad-spectrum antibiotics can also have significant side effects, including gastrointestinal disturbances and an increased risk of infections such as Clostridium difficile. Accurately identifying non-allergic children aids in minimising these adverse outcomes [13,14,15,16]. At the same time, there may be indirect costs such as extended hospital stays and the need for additional treatments due to their side effects, which can further increase expenditure. Appropriate de-labelling may lead to more effective resource allocation for children and their families [17,18].

### 1.3. How Is Penicillin Allergy De-Labelling Undertaken?

Traditionally, testing patients for PCN allergies involves a combination of serum IgE testing, skin prick testing, intradermal testing, patch testing, and drug provocation tests (DPT) with graduated dosing. This remains the appropriate approach in both adult and paediatric patients with histories of severe allergic symptoms. However, it is impractical and unnecessary for all patients with low-risk symptoms to undergo formal allergy testing [19]. Traditional skin prick testing is time-consuming and may have a higher impact on resources than direct drug challenges [20]. It has not been shown to significantly improve the diagnosis of PCN allergy in children when compared with proceeding straight to an oral DPT [21]. Performing an oral DPT without prior allergy tests is now considered the appropriate pathway in children with a history of mild allergic symptoms. It is associated with mild reactions in approximately 5% of cases. Severe reactions are rare (0.036%) and likely related to the risk stratification approach. DPTs take numerous forms, including graded or single observed doses. Extended home challenges lasting 2–5 days are used to explore the history of delayed reactions. The impact of false PCN labelling on individual patient outcomes and societal antimicrobial stewardship are such that it is the responsibility of all healthcare professionals to be involved. Patients with straightforward histories of mild allergic symptoms can be de-labelled by non-allergists. De-labelling beyond the hospital walls, in Primary Care, has been shown to be a safe approach [22]. Moreover, with new technologies using artificial intelligence and telemedicine being added in clinical settings, there is a need to assess their role in improving penicillin de-labelling strategies [23,24]. In this vein, a recent review showed that although telemedicine is a promising alternative in access to penicillin allergy de-labelling, especially in low-risk cases, it cannot substitute an in-person evaluation in mid- and high-risk cases [23].

### 1.4. What Is the Role of the Paediatric Emergency Department?

The Paediatric Emergency Department (PED) offers a unique environment for the rationalisation of PCN allergy labelling. Conversations prior to prescribing antibiotics will commonly reveal parental concerns such as a family history of penicillin allergy or a previous history of a non-immunological adverse reaction to penicillin (diarrhoea). This is an opportunity to reassure that penicillin can be safely re-prescribed. A family history of PCN allergy is neither an indication for allergy testing nor avoidance. In certain cases, a first dose may be delivered in the PED to support parental compliance. Young children commonly present to the PED with antibiotic-related rashes. Parents and carers’ interpretation is generally one of PCN allergy. Validation of this concern by a healthcare professional increases the likelihood of lifelong penicillin avoidance due to parental reluctance for penicillin [12]. The origin of these rashes in 95% of cases is viral exanthem or drug viral interactions (EBV). PED teams supported by appropriate decision tools can stem the tide of PCN allergy labelling. Although clinicians in the PED are unlikely to be allergy specialists, de-labelling performed by non-allergy specialists is supported by meta-analysis data demonstrating this to be a safe and efficacious method of challenging allergy status [15]. The PED offers an environment with a higher skill set, where low-risk de-labelling of children with more challenging comorbidities can be undertaken. The safety of de-labelling actively sick children in the PED requires further research [25].

There are tools trying to discern between those with a high-risk or low-risk allergy status. Some tools, such as PEN-FAST (Table 3), showed excellent negative predictive values, with a cut-off score of 0 demonstrating a <1% risk of demonstrable PCN allergy [4,26,27]. In a recent retrospective cohort study, a cut-off score of 3 resulted in higher sensitivity (100% (95% CI 67.6–100)) and negative predictive value (100% (95% CI 91.2–100)). However, specificity and positive predictive value were low (35% (95% CI 33.8–53.7) and 8% (95% CI 6.9–24.2), respectively) [28]. Such tools also enable the identification of those who are low risk and, therefore, suitable for direct drug challenges without the need for skin prick testing. Randomised controlled trials have demonstrated that such strategies are efficacious in inpatient paediatric populations [20], allowing the de-labelling of 96.2% of patients with this testing, taking 66.7 ± 4.8 min. However, ongoing validation in paediatrics is required [29]. On a pragmatic level, several quality improvement projects have similarly demonstrated success in establishing PCN allergy de-labelling amongst paediatric inpatients [13,30]. A 2021 quality improvement project highlighted crucial multidisciplinary involvement to create an efficient clinical care pathway to support penicillin de-labelling by paediatric hospitalists. This involved targeted workflow optimisation to support de-labelling, establishment of an educational intervention, as well as participation in a hospital’s quality improvement incentive program [30]. In another single-centre RCT in 82 paediatric patients, 2–16 years old, with a parental-reported PCN allergy, 36 out of 37 patients that had PCN de-labelling in the PED were successfully de-labelled. However, the length of stay in the ED was increased in the de-labelling group mainly due to the research consent process and randomization [31]. The success of de-labelling quality improvement projects has also been innovatively expanded to produce open-access educational resources, enabling education and risk assessment in low-resource settings [32]. Whilst in the PED, in a study by the PECARN network, the use of direct oral allergy challenge demonstrated the ability to enable de-labelling of 98% of patients with a parental-reported PCN allergy [33].

Overall, it is recommended that PCN allergy de-labelling should be carried out in two stages. First, it involves an identification phase to screen individuals at low risk of severe reaction. Then, for this group, direct oral allergy challenge and de-labelling may be appropriately offered within the PED [13].

### 1.5. Parent/Caregiver Perceptions on Penicillin Allergy De-Labelling

Frequently, opportunistic intervention in such cases exists in the PED. Understanding the perceptions of parents/caregivers regarding PCN allergy de-labelling is crucial for performing de-labelling in the PED. Clinicians need to have an awareness of parents’ current understanding, openness to testing, and opinions on receipt of antibiotics following successful testing and de-labelling. Both local and international information is available for parents and caregivers to support their understanding [34]. Despite this, many parents and caregivers are unaware that PCN allergy challenge testing exists [34]. Lee et al. demonstrated that 20% of parents and caregivers of children with a documented PCN allergy were aware of challenge testing for PCN allergies [35]. More widespread dissemination and education around PCN allergy de-labelling is a key target for future initiatives to promote this.

The openness of parents/caregivers to their child undergoing direct oral allergy challenge and perceived barriers towards this have been more broadly explored in the literature [12,15,35,36]. Lee et al. described that 55% of parents/caregivers were willing to let their child to undergo PCN allergy testing and de-labelling [35,37]. Amongst the remaining 45% of parents/caregivers who were hesitant, the majority concern was their child having a bad reaction to PCN [35]. Yang et al. described that 59% of parents/caregivers were unwilling to undergo PCN allergy testing and de-labelling during a PED visit [12]. Interestingly, parents/caregivers did not see the need to use alternative antibiotics to PCN as a challenge, issue, or an inconvenience [35]. This has been mirrored amongst the medical community [14,38]. This is a targetable knowledge gap amongst the public and healthcare professionals.

Antoon et al. qualitatively explored the openness of parents/caregivers to PCN allergy challenge testing [36]. Most parents/caregivers felt certain that their child had a previous allergic reaction to PCN. This belief was maintained across repeated healthcare interactions and appeared to be linked to resistance to engagement with de-labelling [36]. Parents/caregivers described feelings of being dismissed by healthcare providers when being told that their child’s symptoms were not allergies and expressed scepticism towards the process of allergy testing and de-labelling [36]. However, discussions with parents/caregivers also revealed key educational points that were useful and helped increase their openness towards testing and de-labelling [36]. For example, many parents/caregivers did not know that the allergy could improve as their child grew, nor were they aware of the negative consequences of inaccurate labelling and the link with antibiotic resistance [36]. This again suggests that education needs to be communicated in a carefully considered manner to create buy-in for testing and de-labelling amongst parents/caregivers [14].

Lee et al. and Yang et al. observed that approximately half of parents/caregivers reported that they would remain uncomfortable with their child receiving PCN even after successful testing and de-labelling [12,35]. The main reasons given were a fear of an allergic reaction and managing this reaction at home. Overall, this suggests that even after de-labelling, there are a number of parents/caregivers who remain reluctant for their child to receive PCN [14]. This number may decline with careful education that is sustained beyond the initial de-labelling event, with normalisation of de-labelling in clinical care [14].

## 2. Conclusions

While many children carry a penicillin allergy label, the vast majority do not have a true immunologically mediated allergy, leading to a lifetime risk of avoidable use of broad-spectrum antibiotics, higher healthcare costs, and poorer clinical outcomes. Despite the challenges posed by barriers such as overcrowding and the need for quick patient turnover in the PED, the availability of resources and expertise in managing potential allergic reactions makes the PED an ideal environment where PCN de-labelling can be both feasible and effective. Further education of parents and caregivers, along with healthcare professionals, is essential to overcoming misconceptions, alleviating safety concerns, fostering trust in the de-labelling process, and normalising de-labelling in the PED.

## Figures and Tables

**Table 1 antibiotics-14-01222-t001:** Drug allergy terminology and definitions.

Terminology	Definition
Adverse drug reaction (ADR)	This is a response to a drug which is noxious and unintended and which occurs at doses normally used in humans for prophylaxis, diagnosis, or therapy of disease or for the modification of physiologic function [9].
Drug allergy	This is defined as an adverse drug reaction with an established immunological mechanism [10].
Type I: immediate-type hypersensitivity	Immediate: <1 h after drug exposure. Anaphylaxis, urticaria, angioedema, gastrointestinal, respiratory, cardiovascular, and neurological symptoms.
Type IV: delayed-type hypersensitivities	Up to 24 h and 1 week after first exposure. IVa (allergic contact dermatitis and maculopapular exanthema), IVb (severe cutaneous skin rashes), IVc (acute generalised pustulosis, DRESS syndrome *, SJS *), IVd (TEN * and generalised bullous FDE *).

* drug reaction with eosinophilia and systemic symptoms (DRESS), Stevens–Johnson syndrome (SJS), toxic epidermal necrolysis (TEN), and fixed drug eruption (FDE).

**Table 2 antibiotics-14-01222-t002:** Risk classification for allergies.

High-Risk Allergy Symptoms	Low-Risk Allergy Symptoms	Not Immune-Mediated
Immediate cutaneous symptoms (within 6 h of ingestion; typically 1 h) Anaphylaxis Orofacial swelling Severe cutaneous adverse reactions	Delayed cutaneous symptoms (maculopapular exanthem, urticaria, pruritus) Patient does not recall allergy No hospital intervention	Isolated gastrointestinal symptoms (nausea, vomiting, diarrhoea) Neurological symptoms (headache and mood disorders) Family history of penicillin allergy

**Table 3 antibiotics-14-01222-t003:** PEN-FAST clinical decision rule for point-of-care risk assessment of patient-reported penicillin allergies.

Question	Score
**PEN**—Penicillin allergy reported by patient	Yes = Use PEN-FAST No = PEN-FAST not applicable
**F**—Five years or less since reaction	Yes = +2 No = 0
**A**—Anaphylaxis or angioedema	Yes = +2 No = 0
**S**—Severe cutaneous adverse reaction	Yes = +1 No = 0
**T**—Treatment required for reaction	Yes = 1 No = 0

0 points = Very low risk of positive penicillin allergy test <1%; 1–2 points = low risk of positive penicillin allergy test 5%; 3 points = moderate risk of positive penicillin allergy test 20%; 4 points = high risk of positive penicillin allergy test 50%.

## Data Availability

No new data were created or analyzed in this study. Data sharing is not applicable to this article.

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
