# Peer review of "De-Labelling Penicillin Allergies in the Paediatric Emergency Department"

_antibiotics, 2025, doi:10.3390/antibiotics14121222_

Round 1
Reviewer 1 Report
Comments and Suggestions for Authors
An interesting review about penicillin allergy in the paed emergency Dept. It highlights how most children with reported penicillin allergy are not actually allergic.
The most important section of the paper is in lines 148-162, where the authors cite papers that provide evidence for their suggestion. This section needs expanding providing more details of the actual studies ,eg number of patients, which health professionals were involved, were additional resources needed to carry out testing, etc.
Providing this detailed evidence would strengthen their paper considerably
Author Response
An interesting review about penicillin allergy in the paed emergency Dept. It highlights how most children with reported penicillin allergy are not actually allergic.
The most important section of the paper is in lines 148-162, where the authors cite papers that provide evidence for their suggestion. This section needs expanding providing more details of the actual studies, e.g. number of patients, which health professionals were involved, were additional resources needed to carry out testing, etc.
Providing this detailed evidence would strengthen their paper considerably
We thank the reviewer for the edits.
We revised manuscript and now reads as follows:
Lines 157-160:
In a recent retrospective cohort study, a cut-off score of 3 resulted in higher sensitivity (100% (95% CI 67,6–100)) and negative predictive value (100% (95% CI 91,2–100)).However specificity and positive predictive value were low (35% (95% CI 33,8–53,7) and 8% (95% CI 6,9–24,2) respectively) [29].
Lines 167-175:
A 2021 quality improvement project highlighted crucial multidisciplinary involvement to create an efficient clinical care pathway to support penicillin de-labeling by pediatric hospitalists. This involved targeted workflow optimization to support de-labeling, establishment of an educational intervention as well as participation in hospital’s quality improvement incentive program [31] In another single center RCT in 82 pediatric patients 2-16-years-old with parental reported PCN allergy, 36 out of 37 patients that had PCN de-labeling in the PED were successfully de-labeled. However, length of stay in the ED was increased in the de-labeling group mainly due to research consent process and randomization [32].
Reviewer 2 Report
Comments and Suggestions for Authors
Dear colleagues thank you for your interesting review regarding penicillin de-labelling in pediatric patients.
In rows 148-162, page 4, you are describing PEN-FAST clinical decision rule for selecting patients that could undergo DPT [direct provocation test].
And you are stating that such a tool "with a cut-off score of 0 has demonstrated a <1% risk of demonstrable PCN allergy" [row 150].
Maybe you could nuance this cut-off rather extreme value with a different cut-off value with a more real-life option?
A value of 3, as stated by a recent 2025 paper is documenting a more optimistic value (Yağmur H, et al. Evaluation of drug provocation tests without prior skin testing in children with suspected penicillin allergy and correlation with PEN-FAST: A single-center study. Eur J Pediatr. 2025 Jul 17;184(8):488. doi: 10.1007/s00431-025-06301-7. PMID: 40676330; PMCID: PMC12271276.), is stating:
"when a PEN-FAST score ≥ 3 was used as the cutoff, the sensitivity of the PEN-FAST test in identifying true drug allergy based on DPT results was calculated as 100% (95% CI 67,6–100), specificity as 35% (95% CI 33,8–53,7), positive predictive value (PPV) as 8% (95% CI 6,9–24,2), and negative predictive value as 100% (95% CI 91,2–100)."
Recent impact of COVID-19 pandemic documented the need for change in addressability and accessibility for a broad set of medical tools and interventions.
Among such interventions that could be delivered remotely, via telemedicine, without physical presence in hospital has been included also a selected group of patients that had penicillin allergy de-labeling (Smolinska S, et al. Mhealth. 2025 Oct 28;11:66. doi: 10.21037/mhealth-24-89. PMID: 41211001; PMCID: PMC12594011).
Authors are concluding with caution that "while TM offers promising avenues to expand access to penicillin allergy de-labeling in low-risk individuals, it is not a substitute for in-person evaluation in high- or intermediate-risk cases".
New tools like "AI chatbots, can significantly enhance clinical practice in allergy and immunology" as stated by a relative recent review (Goktas P, et al. Artificial Intelligence Chatbots in Allergy and Immunology Practice: Where Have We Been and Where Are We Going? J Allergy Clin Immunol Pract. 2023 Sep;11(9):2697-2700. doi: 10.1016/j.jaip.2023.05.042. Epub 2023 Jun 8. PMID: 37301435.)
Could you include a comment on these new tools, that integrate AI-tools and new protocols in penicillin de-labeling?
In row 154 you should correct the typo error that presents reference 21 in an abnormal format.
Author Response
Dear colleagues thank you for your interesting review regarding penicillin de-labelling in pediatric patients.
We thank the reviewer for the feedback and positive comments.
In rows 148-162, page 4, you are describing PEN-FAST clinical decision rule for selecting patients that could undergo DPT [direct provocation test].
And you are stating that such a tool "with a cut-off score of 0 has demonstrated a <1% risk of demonstrable PCN allergy" [row 150].
Maybe you could nuance this cut-off rather extreme value with a different cut-off value with a more real-life option?
A value of 3, as stated by a recent 2025 paper is documenting a more optimistic value (Yağmur H, et al. Evaluation of drug provocation tests without prior skin testing in children with suspected penicillin allergy and correlation with PEN-FAST: A single-center study. Eur J Pediatr. 2025 Jul 17;184(8):488. doi: 10.1007/s00431-025-06301-7. PMID: 40676330; PMCID: PMC12271276.), is stating:
"when a PEN-FAST score ≥ 3 was used as the cutoff, the sensitivity of the PEN-FAST test in identifying true drug allergy based on DPT results was calculated as 100% (95% CI 67,6–100), specificity as 35% (95% CI 33,8–53,7), positive predictive value (PPV) as 8% (95% CI 6,9–24,2), and negative predictive value as 100% (95% CI 91,2–100)."
Thank you for the suggestion – we revised the manuscript and now reads as follows:
Lines 157-160: In a recent retrospective cohort study, a cut-off score of 3 resulted in higher sensitivity (100% (95% CI 67,6–100)) and negative predictive value (100% (95% CI 91,2–100)).However specificity and positive predictive value were low (35% (95% CI 33,8–53,7) and 8% (95% CI 6,9–24,2) respectively) [27].
Recent impact of COVID-19 pandemic documented the need for change in addressability and accessibility for a broad set of medical tools and interventions.
Among such interventions that could be delivered remotely, via telemedicine, without physical presence in hospital has been included also a selected group of patients that had penicillin allergy de-labeling (Smolinska S, et al. Mhealth. 2025 Oct 28;11:66. doi: 10.21037/mhealth-24-89. PMID: 41211001; PMCID: PMC12594011).
Authors are concluding with caution that "while TM offers promising avenues to expand access to penicillin allergy de-labeling in low-risk individuals, it is not a substitute for in-person evaluation in high- or intermediate-risk cases".
New tools like "AI chatbots, can significantly enhance clinical practice in allergy and immunology" as stated by a relative recent review (Goktas P, et al. Artificial Intelligence Chatbots in Allergy and Immunology Practice: Where Have We Been and Where Are We Going? J Allergy Clin Immunol Pract. 2023 Sep;11(9):2697-2700. doi: 10.1016/j.jaip.2023.05.042. Epub 2023 Jun 8. PMID: 37301435.)
Could you include a comment on these new tools, that integrate AI-tools and new protocols in penicillin de-labeling?
Thank you for the suggestion. Revised and now reads as follows:
Lines 128-133:
Moreover, with new technologies using artificial intelligence and telemedicine being added in the clinical settings, there is a need to assess their role in improving penicillin de-labelling strategies [24,25]. In this vein, a recent review showed that although tele-medicine is a promising alternative in access to penicillin allergy de-labeling especially in low-risk cases, it cannot substitute an in-person evaluation in mid- and high-risk cases [24].
In row 154 you should correct the typo error that presents reference 21 in an abnormal format.
Revised- thanks for noticing.